# The dominant–egalitarian transition in species-rich communities

David A Kessler*, Nadav M Shnerb*

Department of Physics, Bar-Ilan University, Ramat-Gan, Ramat-Gan, Israel

**Abstract** Diverse communities of competing species are generally characterized by substantial niche overlap and strongly stochastic dynamics. Abundance fluctuations are proportional to population size, so the dynamics of rare populations is slower. Hence, once a population becomes rare, its abundance gets stuck at low values. Here, we analyze the effect of this phenomenon on community structure. We identify two distinct phases: a dominance phase, in which a tiny number of species constitute most of the community, and an egalitarian phase, where it takes a finite fraction of all species to constitute most of the community. Using empirical data from microbial, planktonic, and macroorganismal systems, we demonstrate the relevance of this transition and show how demographic stochasticity and immigration critically determine phase behavior. Our results suggest that even slight changes in noise strength or immigration rates can lead to abrupt shifts in community diversity.

## Editor's evaluation

This valuable paper explores the question of when complex ecosystems will be dominated by a few species. The authors present compelling, general arguments for a phase transition from what they call a dominance phase (few species dominate biomass) to an egalitarian phase (no species dominates the biomass).

**\*For correspondence:**
David.Kessler@biu.ac.il (DAK);
nadav.shnerb@gmail.com (NMS)

**Competing interest:** The authors declare that no competing interests exist.

## Introduction

Many ecological systems contain numerous competing species, strains, or types (*Hutchinson, 1961*; *Stomp et al., 2011*; *ter Steege et al., 2013*; *Connolly et al., 2014*; *Fierer et al., 2007*). In such systems, it can be assumed a priori that the structure of the community essentially reflects resource competition among species (*Gause, 2003*; *Tilman, 1982*), with factors such as niche overlap and fitness differences playing a central role in the community assembly (*Chesson, 2000*; *Chesson, 2003*). Analyzing these factors and their impact is crucial for understanding the dynamics of these systems, and consequently, for our ability to intervene in these dynamics to achieve desired outcomes – from maintaining biodiversity in a changing world to successfully altering the state of the gut microbiome (*David et al., 2014*; *Grilli, 2020*; *Eguíluz et al., 2019*; *Cooper et al., 2024*; *Callaghan et al., 2021*).

Unfortunately, progress in this area has been quite difficult. The coexistence of many species remains largely puzzling, especially given the competitive exclusion principle (*Tilman, 1982*) and the strict constraints on the stability and feasibility of such complex systems (*May, 1972*). Moreover, quantifying the relevant parameters in diverse communities is extremely challenging, particularly considering the high level of stochasticity typically present in ecological dynamics. As a result, approaches inspired by statistical physics, which examine generic models through a few summary statistics, have gained significant popularity in recent years (*Fisher and Mehta, 2014*; *Kessler and Shnerb, 2015*; *Bunin, 2017*; *Barbier et al., 2018*; *Grilli, 2020*; *van Nes et al., 2024*).

Broadly speaking, attempts to present a generic analysis of diverse communities can be divided into two main approaches that differ in their interpretation of what determines dominance. In one class

of models, the identity of the high-abundance species reflects intrinsic fitness differences (whether through growth rates, competitive advantage, or resource-use efficiency) so that the dominant species are effectively the 'fittest' (*Fisher and Mehta, 2014*; *Bunin, 2017*; *Barbier et al., 2018*; *Azaele and Maritan, 2023*; *Marcus et al., 2022*). In contrast, an alternative view holds that dominance arises from contingent stochastic dynamics: species happen to be abundant not because they are intrinsically superior, but as a result of random historical trajectories. These trajectories may reflect demographic noise, environmental stochasticity, or even deterministic chaos in systems with complex interactions.

The prototype of this second class of approaches was introduced in the neutral model proposed by *Crow and Kimura, 1970* and *Hubbell, 2001*. In the original neutral models, demographic stochasticity – the inherent randomness in the birth–death process at the individual level – is the sole driver of abundance variations. Analytical solutions to these models are relatively easy to obtain (*Volkov et al., 2003*; *Azaele et al., 2015*) and have been quite successful in explaining the observed species abundance distributions (SADs), as well as other static patterns, in both regional and local communities (*Rosindell et al., 2011*). However, pure demographic stochasticity cannot account for dynamic patterns, whether evolutionary (such as the time to the most recent common ancestor) (*Nee, 2005*; *Ricklefs, 2006*) or ecological (such as the dynamics of abundance variation and similarity indices) (*Leigh, 2025*; *Chisholm and O'Dwyer, 2014*; *Kalyuzhny et al., 2014b*; *Kalyuzhny et al., 2014a*). Demographic stochasticity results in relatively slow and weak abundance variations, whereas observed variability is much stronger and occurs more rapidly (*Kalyuzhny et al., 2014b*; *Chisholm et al., 2014*).

The time-averaged neutral model (*Kalyuzhny et al., 2015*) addresses these limitations by relaxing the assumption of time-independent fitness. In this model, the relative fitness of each species fluctuates over time, but all species share the same mean fitness. The analysis of the time-averaged neutral model is more complex, as environmental stochasticity can facilitate coexistence through the storage effect (*Chesson and Warner, 1981*). However, the significance of this effect diminishes in highly diverse systems (*Dean et al., 2017*; *Danino and Shnerb, 2018*; *Pande and Shnerb, 2022*; *Meyer et al., 2022*).

Recent studies have highlighted a notable effect of environmental stochasticity in competitive communities, known as 'stickiness' (*van Nes et al., 2024*) or 'diffusive trapping' (*Dean and Shnerb, 2020*). Under environmental stochasticity, abundance variations are proportional to population size, meaning that the dynamics of rare species is slow, causing them to linger near the extinction point for extended periods. *van Nes et al., 2024* suggested that this stickiness enables time-averaged neutral models to produce patterns – such as changes in abundance over time, dominant species turnover, and community egality – that closely resemble those observed in real ecological communities. Similar results were found by *Mallmin et al., 2024* in a system of competing species for which the overall dynamics are chaotic. This makes sense, given that the relative fitness of a specific species depends on the abundances of its competitors, so if these abundances fluctuate strongly over time, as they do in the chaotic phase, the community will likely reach a state that resembles time-averaged neutrality.

Our main goal in this article is to identify a previously unrecognized phase transition between dominant and egalitarian communities. To describe this transition, we define the number of dominant species, $S_{1/2}$, as the minimum number of species that must be grouped together at a given moment to account for more than half of the total population. The level of equality in the community is then measured by the egality ratio, $S_{1/2}/S$, where $S$ is the total species richness. We show that in the dominance phase, the growth of $S_{1/2}$ with $S$ is sublinear, so the egality parameter approaches zero as $S \to \infty$, whereas in the egalitarian phase, $S_{1/2}$ grows linearly with $S$, and the egality parameter converges to a finite value. We demonstrate the existence of these two phases through analysis of empirical data on patterns of commonness and rarity across a wide range of ecological systems, from tropical forest trees to the human microbiome.

During the analysis and comparisons with the empirical data, we also arrived at two important insights.

- The basic model presented by *van Nes et al., 2024* includes only growth, competition, and environmental stochasticity. We show that this dynamics alone leads to monodominance, where all populations except one decline below any finite abundance level. Coexistence requires a mechanism, such as immigration, that imposes a lower bound on species abundances. Therefore, immigration – which was considered only in the supplementary materials of *van Nes et al., 2024* – is a crucial ingredient, and so even small changes in the immigration rate can shift the system between ecological phases.

- An additional factor not included in the model of **van Nes et al., 2024** is demographic stochasticity. We will show that for populations of macroorganisms, the abundance distribution of relatively rare species is significantly affected by demographic stochasticity, and to properly describe it, this mechanism must also be taken into account.

The dominant–egalitarian phase transition has practical implications. A small decrease in the immigration rate or, alternatively, a small increase in the level of stochasticity can lead to a sharp decline of the diversity within a given community. Moreover, in the section 'Materials and methods', we have examined cases in which neutrality is broken, including fitness differences and asymmetry in the competitive interaction matrix. The results of this study demonstrate that the phase transition between the egalitarian and dominance regimes is not an artifact of perfect neutrality, but remains robust over a finite region of parameter space near the neutral point.

## Results

As we have already mentioned, without external immigration, the stickiness will eventually cause the abundance of all species, except one, to drop below any finite threshold, leading the community to a state of monodominance. To avoid disrupting the continuity of the main discussion, we detail and demonstrate this claim in 'SI A' (see also the supplement to **Figure 1**). Therefore, the basic model we study is given by

$$
\begin{aligned}
\frac{dN_i}{dt} &= \mu + N_i \left( 1 - \frac{\sum_{j=1}^{S} N_j}{K} \right) + r_i(t)N_i \\
\frac{dr_i}{dt} &= -\frac{r_i}{\tau} + \theta \eta_i(t),
\end{aligned}
\tag{1}
$$

where $N_i$, $i = 1, \ldots, S$ is the abundance of the $i$th species. $K$ sets the overall carrying capacity. The growth rate of the $i$th species is $1 + r_i(t)$, where $r_i(t)$ is an independently fluctuating variable, with a

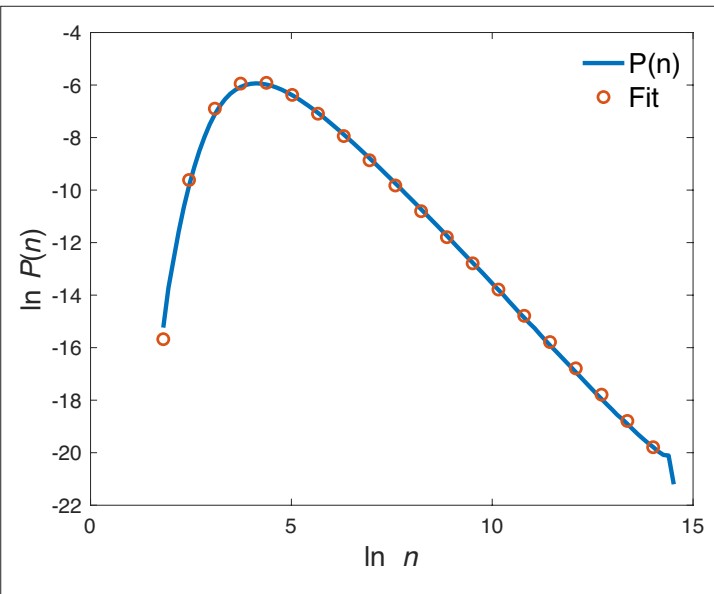

**Figure 1.** Probability distribution of species abundance. Red circles: results of a simulation of the process described in **Equation 1** with parameters $S = 200$, $\mu = 0.6$, $K = 2 \cdot 10^6$, $\tilde{\sigma}_e = 0.1$ and $\tau = 8$. The values of the extracted parameters for **Equation 4** are $c = 0.0034$ and $\tilde{\sigma}_e^2 = 0.0131$. The solid line in blue is the expression of **Equation 4** with parameters $\alpha = 91.2$ and $\beta = 1.56$.

The online version of this article includes the following figure supplement(s) for figure 1:

**Figure supplement 1.** Species abundances (left), log-abundance (middle), and the egality parameter $S_{1/2}/S$ (right) for the model considered in the main text of **van Nes et al., 2024**, Parameters are $S = 100$, $\tilde{\sigma}_e = 0.005$ and $K = 100$.

correlation time $\tau$, driven by the white noise $\eta_i$. The immigration rates of all species are identical and given by $\mu > 0$. Further details on the model are given in the 'Materials and methods' section.

is an independently fluctuating variable, with a correlation time $\tau$, driven by the white noise $\eta_i$. The immigration rates of all species are identical and given by $\mu > 0$. Further details on the model are given in the 'Materials and methods' section.

## Stickiness and species abundance distribution

Without immigration, the carrying capacity parameter $K$ plays no role in the dynamics. Actually, by defining $N_i = Kn_i$ one may rescale *Equation 1* to $K = 1$. When immigration is introduced, the parameter $K/\mu$ expresses the ratio between the carrying capacity and the typical minimum level of population size. The larger this ratio, the stronger the stickiness effect.

On top of that, since stickiness arises from environmental stochasticity, it becomes stronger as $\tilde{\sigma}_e^2 \equiv \theta^2 \tau^2$ increases. Dimensional analysis reveals that the dimensionless parameter that governs stickiness is

$$\gamma \equiv \frac{K\tilde{\sigma}_e^2}{\mu}.$$

The larger $\gamma$ is, the stronger is the stickiness, and the community approaches monodominance when $\gamma \to \infty$; for example, when $\mu \to 0$.

In order to advance in the analysis, we are interested in replacing *Equation 1*, which provides us with a description of an $S$-dimensional system where each species can affect every other species, with an effective, one-dimensional equation for a focal species, considering all others as a single rival species. In the classical neutral theory, with pure demographic stochasticity, this can be done trivially, as the species identity of a particular individual plays no role in the dynamics. In the time-averaged neutral theory, however, the situation is much more subtle.

The distinction between neutral and time-averaged neutral models – as it arises in the attempt to derive an effective one-species description – was clarified in *Danino and Shnerb, 2018* and *Steinmetz et al., 2020*, and has to do with the difference between the time-averaged growth rate of a given species and the population-averaged growth rate of the community. In *Equation 1*, the linear growth rate of each species is $1 + r(t)$. Since the $r$ process is symmetric around zero, the time-average growth rate of each species is unity. Nevertheless, the instantaneous growth rate of the *community* is, on average, greater than 1. At any given moment, the fitter species are growing faster, so that on average more than 50% of the individuals belong to instantaneously beneficial species. As a result, the typical value of $\sum_j N_j > K$, and therefore the dynamics of a single species satisfies the effective one-dimensional stochastic differential equation,

$$\frac{dN}{dt} = \mu + \left( \frac{\sigma_e^2}{2} - c \right) N + \sigma_e \eta(t) N, \tag{2}$$

where $c = \mathrm{E}[(\sum_j N_j - K)/K]$. The noise in the net growth rate of the $i$th species arises from two sources, the first being the direct effect of $\eta_i(t)$, and the second being the fluctuations in the competition term, $-\sum_j N_j/K$. The two parameters $c$ and $\sigma_e$ may be measured, for any given values of $K$, $S$, and $\tilde{\sigma}_e$, from long simulations of *Equation 1*. Unless we are looking at the most abundant species, the correlations between the competition term and $\eta_i$ are small, and the two contributions are roughly independent. Thus, the overall effect of these two contributions is

$$\sigma_e^2 \approx \tilde{\sigma}_e^2 + \sigma_c^2 \tag{3}$$

Here, $\sigma_c^2$ is the time integral of the two-time correlation function of $c$ and equals $2\tau \mathrm{Var}[\sum N_i/K]$. For example, for $\tilde{\sigma}_e = 0.01$, $S = 200$, $K = 2 \cdot 10^6$, $\mu = 0.6$, we have $\sigma_c^2 \doteq 0.0031$ and $\sigma_e^2 \doteq 0.1031$, in line with *Equation 3*. The Stratonovich term $\sigma_e^2/2$ expresses the fact that for a population that sometimes grows exponentially and sometimes declines, even if its average growth rate is zero, the arithmetic mean still increases over time. Once this term is introduced, the standard Ito calculus may be applied to *Equation 2*.

Once the relevant parameters are calibrated, the distribution for $P(N)$ may be extracted from *Equation 2*, see 'SI 2'. The resulting distribution is

$$P(n) = Ae^{-\alpha/n}n^{-\beta}, \tag{4}$$

where $\alpha = 2\mu/\sigma_e^2$, $\beta = 1 + 2c/\sigma_e^2$ and $A$ is a normalization factor. *Figure 1* illustrates the success of the approximation and the validity of *Equation 4* in a simulation of a community of $S = 200$ interacting species governed by *Equation 1*.

A similar result was presented a few months ago by *Mallmin et al., 2024*, who dealt with a system of competing species where the community dynamics is chaotic (but without external stochasticity). In such a case, one can consider, for each focal species, all other species as an effective external environment whose fluctuations generate stochasticity in the instantaneous growth rate of the focal species. A discussion of the similarities and differences between our case and the chaotic model will be presented below.

## The egalitarian transition

The most striking implication of *Equation 4* is the sharp shift in the compositional properties of the community at $\beta = 2$. When $\beta > 2$, abundant species are relatively rare, resulting in an 'egalitarian' community. Conversely, when $\beta < 2$, the community composition is dominated by a few exceptionally abundant species.

To quantify the egality of the community, one may use the criterion suggested by *van Nes et al., 2024*, which involves comparing the total number of species, $S$, to the minimal number of species required to make up half of the community, $S_{1/2}$. In a more egalitarian community, the fraction $S_{1/2}/S$ is finite, indicating that the number of dominant species is proportional to the total number of species. In contrast, in a community dominated by only a few species, the ratio $S_{1/2}/S$ approaches zero as $S$ increases, meaning the number of dominant species is sublinear in $S$.

In Appendix C, we provide the relevant mathematical analysis, demonstrating that, as $\beta$ decreases towards 2, $S_{1/2}/S$ monotonically decreases, and in the limit $\beta \to 2^+$,

$$\frac{S_{1/2}}{S} \sim e^{-(\ln 2)/(\beta - 2)}. \tag{5}$$

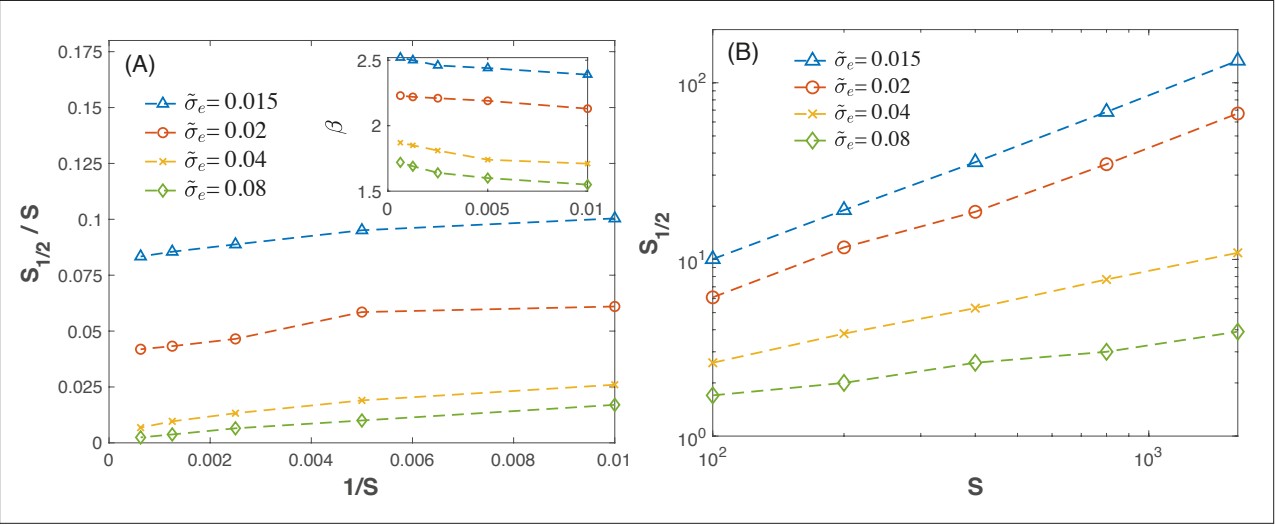

**Figure 2.** The egality parameter as a function of the number of species. Left panel (**A**): the egality parameter $S_{1/2}/S$ is plotted against $1/S$, the total number of species, for $\tilde{\sigma}_e = 0.015$ (blue) 0.02 (red) 0.04 (orange) and 0.08 (green). For all these simulations $\mu = 0.6$, $\tau = 8$ and $K = 10^4$. The values of the parameter $\beta$, as extracted from the simulation, are plotted (again, vs. $1/S$) in the inset. As expected, for $\beta > 2$ (blue) the egality parameter $S_{1/2}/S$ extrapolates, at $S \to \infty$, to a finite value, meaning that the fraction of high-abundance species is finite and the community is egalitarian. This feature is also reflected in the right panel (**B**), where at large $S$, $S_{1/2} \sim 0.04S$ for $\tilde{\sigma}_e = 0.02$ (red) and $S_{1/2} \sim 0.08S$ for $\tilde{\sigma}_e = 0.015$ (blue). On the other hand, when $\tilde{\sigma}_e = 0.08$ (green), the value of $\beta$ approaches 1.8 and is definitely smaller than two. In addition, the egality parameter extrapolates to zero at large $S$, meaning that the community is dominated by a small number of species. The results shown in the right panel suggest $S_{1/2} \sim S^{0.3}$. The case $\tilde{\sigma}_e = 0.04$ (orange) represents a near marginal case, for which $\beta$ extrapolates to values slightly smaller than 2, and it would appear that $S_{1/2}/S$ is tending toward zero. Indeed, the right panel shows a power law with exponent smaller than unity and decreasing with increasing $\tilde{\sigma}_e$.

Thus, this egality parameter vanishes in a singular manner in that limit, indicating the transition out of the egalitarian phase.

The numerical results presented in *Figure 2A* illustrate this phenomenon and offer several additional interesting insights. Beyond the positive indication regarding the result in the limit where $S \to \infty$, *Figure 2A* shows that the value of $\beta$ depends only weakly on $S$ so it can be considered approximately constant. Moreover, as seen in *Figure 2B*, for $\beta < 2$ the dependence of $S_{1/2}$ on $S$ follows a power law, with an exponent approaching unity from below at the phase transition point $\beta = 2$.

Of course, there are other quantities one could employ to quantify the egality. One such possibility is the inverse participation ratio, $I$, defined as $I = \left[\sum_i (N_i/K)^2\right]^{-1}$. This varies from 1 in the limit of single species dominance to $S$ for an egalitarian community. Our measurements (not shown) indicate that, similarly to $S_{1/2}$, $I$ increases with $S$, though the exponents controlling the growth are slightly lower. For example, for $\tilde{\sigma}_e = 0.08$, $I \sim S^{0.26}$, as opposed to $S_{1/2} \sim S^{0.29}$, and for $\tilde{\sigma}_e = 0.4$, $I \sim S^{0.4}$, while $S_{1/2} \sim S^{.52}$.

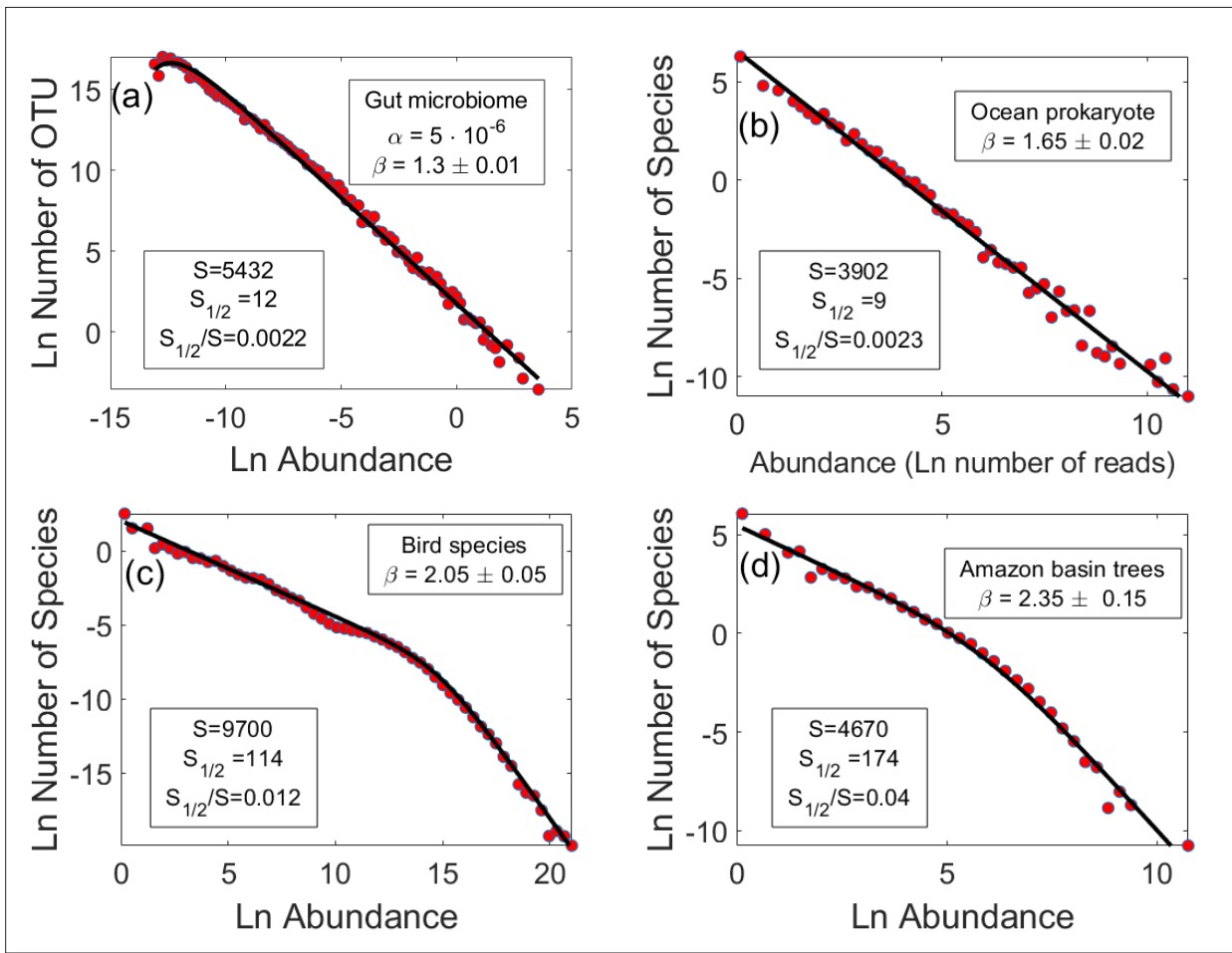

**Figure 3.** Species abundance distributions, as obtained for a few diverse communities. The gut microbial community (OTUs of subject A from *David et al., 2014*) fits our formula *Equation 4* quite well. Immigration is extremely weak, but the sampling power is strong enough to reveal some of the effects of the decrease in the number of species at low densities due to immigration. The corresponding distribution for the oceanic prokaryote population *Eguíluz et al., 2019* is very close to a pure power law, possibly because the sampling is not deep enough (see text). In both cases, $\beta < 2$, indicating that the community is dominated by only a few species. The distributions for the global bird population (*Callaghan et al., 2021*) and tropical trees in the Amazon Basin (*Cooper et al., 2024*) show a clear transition between two power-law behaviors. As explained in the text, for these macroorganisms the effect of demographic stochasticity must be taken into account. When this is done (see 'SI D'), we obtain an excellent fit for the results using the corrected formula. *Equation 6*. For birds, we find $\beta \approx 2$, placing them on the margin between egalitarian and dominance communities. For tropical trees in the Amazon Basin, $\beta$ is definitely larger than two, indicating that the community is indeed egalitarian. Note the similarities between the values of $S_{1/2}/S$ in the empirical results and the numerical experiments in *Figure 2*. For more information, see 'SI E'.

## Comparison with empirical results and the impact of demographic stochasticity

Let us now examine the SAD in several cases of diverse communities, assess their degree of alignment with the results presented above, and attempt to differentiate between dominance and egalitarian communities, linking the results to the fundamental characteristics of each system.

We analyze the community structure in four systems: the human gut microbiome (**David et al., 2014**), marine prokaryotes (**Eguíluz et al., 2019**), tropical trees (**Cooper et al., 2024**), and bird species (**Callaghan et al., 2021**). All of these communities are hyperdiverse, with thousands of species, making them reflective of the limit where $1/S$ is very close to zero – a limit where the distinction between egalitarian and dominance systems is clearly defined, with the relevant values of $S_{1/2}/S$ being those shown in **Figure 2**.

**Figure 3a** shows the excellent agreement between the empirical data for the gut microbiome and our model prediction (**Equation 4**). In the case of ocean prokaryotes, presented in panel (b), only a pure power law is observed, perhaps because the sampling strength is insufficient. Weak sampling shifts the distribution leftward, thus hiding the true characteristics of the SAD for small abundance populations (**Maruvka et al., 2010**). In both cases $\beta < 2$, indicating that these systems are in the dominant phase, consistent with the very small values of $S_{1/2}/S$ observed in the data.

In contrast, in **Figure 3c and d**, for birds and Amazon basin trees, respectively, the empirical distribution does not fit **Equation 4**. Instead, the population size distribution shows a pronounced crossover between two descending power laws, without a maximum point at a finite abundance.

To explain this, we note that trees and birds differ from microorganisms in two important ways. First, since macroorganisms tend to have larger body sizes and longer lifespans, their metabolism buffers them against environmental fluctuations. Additionally, since populations of macroorganisms often span wide geographic ranges, most environmental variations appear uncorrelated across populations. Environmental variation that affects individuals, or small groups within the population, in an uncorrelated manner contributes to demographic stochasticity rather than to environmental stochasticity, which, by definition, affects entire populations coherently. Therefore, for macroorganisms we need to account for demographic stochasticity.

In 'SI D', we consider the case of a time-averaged neutral community with demographic stochasticity in addition to the previously considered immigration and environmental stochasticity. The expected distribution for species abundances now takes the form

$$P(n) = An^{-1+2\mu/\sigma_d^2} \left( \frac{\sigma_d^2}{\sigma_e^2} + n \right)^{-2c/\sigma_e^2 - 2\mu/\sigma_d^2}, \tag{6}$$

with $\sigma_d$ parameterizing the strength of demographic stochasticity. **Equation 6** converges to **Equation 4** for values of $n$ satisfying $n \gg \sigma_d^2/\sigma_e^2$ and predicts a different power-law for smaller values of $n$. With this new formula, one can now successfully fit the empirical data in **Figure 3c and d**. It is noteworthy that the transition point between the two power-laws happens for values of the abundance which are quite large, reinforcing our above comments about the effect of the large spatial and temporal scales of macroorganisms on reducing $\sigma_e$ and increasing $\sigma_d$.

Even with the inclusion of demographic stochasticity, the distinction between dominant and egalitarian communities depends solely on the decay of the tail and its corresponding exponent $\beta$, which, for **Equation 6**, remains as $\beta = -1 - 2c/\sigma_e^2$. **Figure 3** thus illustrates the three types of behavior demonstrated in the numerical experiments shown in **Figure 2**: the microorganism communities are in the dominance phase, the tropical tree community is in the egalitarian phase, and the bird community appears to fall in between.

A comparison between the values of the egality parameter in **Figure 2** and the values of $S_{1/2}/S$ in **van Nes et al., 2024** (for the same range of species richness $S$) also shows that some communities fall within the dominance phase, while others fall within the egalitarian phase. On the other hand, the results for plankton (**Ser-Giacomi et al., 2018**), in which $\beta \in [1...2]$, appear to suggest that these microorganismal communities are all in the dominance phase.

## Relaxing the assumption of neutrality

So far, we have analyzed the properties of a system that is fully neutral in the deterministic limit – that is, all species have exactly the same fitness. In this section, we aim to explore, or at least provide

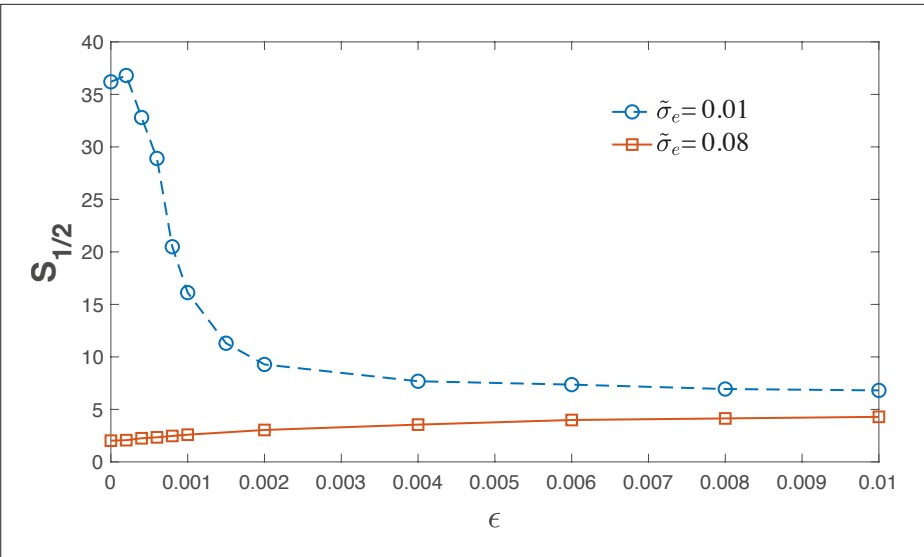

**Figure 4.** From stochasticity-driven to chaotic dynamics. $S_{1/2}$, as obtained from simulations of **Equation 7**, is plotted against $\epsilon$, for $N = 200$, $K = 2 \cdot 10^6$ and $\mu = 0.6$. For $\tilde{\sigma}_e = 0.01$ (blue circles), the system is in the egalitarian phase and the main effect of the transition to chaos is an effective enhancement of the strength of stochasticity that makes the system less egalitarian, hence $S_{1/2}$ decreases. The inverse effect is observed for $\sigma_e = 0.08$ (red squares). For large values of $\epsilon$, the effect of $\sigma_e$ becomes negligible and the dependence of $S_{1/2}$ on $\sigma_e$ weakens significantly. The data was obtained by averaging over at least 20 different realizations of the interaction matrix $\alpha_{i,j}$.

The online version of this article includes the following figure supplement(s) for figure 4:

**Figure supplement 1.** Time traces of species abundances for $\epsilon = 0$ (left panel, time-averaged neutral) and $\epsilon = 0.05$ (right panel, chaotic), with $S = 200$, $K = 10^4 S$, and $\tau = 8$.

a glimpse of, the outcomes that arise when this assumption is relaxed; in particular, we assess the robustness of our analysis in the vicinity of the neutral point.

To break perfect neutrality, we introduce heterogeneity into the interaction matrix. Specifically, we replace **Equation 1** by

$$\frac{dN_i}{dt} = \mu + N_i \left( 1 - \frac{N_i + \sum_{j \neq i}(1 - 3\epsilon + \epsilon\alpha_{i,j})N_j}{K} \right) + r_i(t)N_i$$

$$\frac{dr_i}{dt} = -\frac{r_i}{\tau} + \theta\eta_i(t),$$

(7)

where $\alpha_{i,j}$ are zero-mean, unit-variance Gaussian random numbers, and $\alpha_{i,j}$ is independent of $\alpha_{j,i}$, so the interaction matrix is asymmetric. For $\epsilon = 0$, this reduces to our original time-averaged neutral model. The additional $-3\epsilon$ term ensures that species compete most strongly against themselves, as is commonly assumed in ecological models, since the niche overlap between individuals of the same species is maximal.

The results are presented in **Figure 4**, which shows $S_{1/2}$ as a function of $\epsilon$ for two different values of the environmental stochasticity $\tilde{\sigma}_e$. As $\epsilon \to 0$, the system may reside either in the egalitarian phase (if $\tilde{\sigma}_e$ is small) or in the dominance phase under strong environmental stochasticity. In contrast, for large values of $\epsilon$, $S_{1/2}$ becomes independent of $\tilde{\sigma}_e$.

To understand these results, note that in the *absence* of environmental stochasticity, the heterogeneous and asymmetric interactions lead to chaotic dynamics of the trajectories, as discussed in **Arnoulx de Pirey and Bunin, 2024**. Illustrations of fluctuation dynamics dominated by stochasticity versus those dominated by deterministic chaos are provided in 'SI F'. Under chaotic dynamics, the system supports, at any instant of time, a clique of abundant species. This clique is *stable* in the sense that if all the species within it are placed together in a community, they will reach an equilibrium state in which all of them persist. However, the clique is *invadable*, meaning that at any given moment, there exist species outside the clique that are capable of invading and reaching high abundance

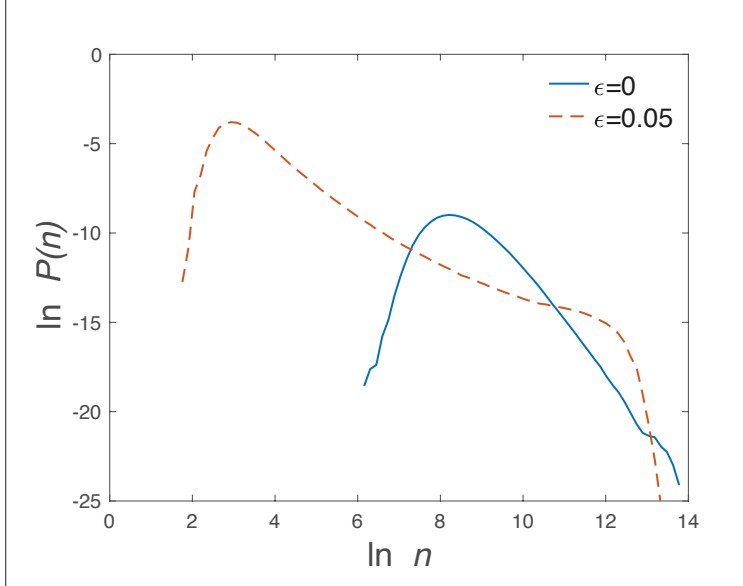

**Figure 5.** $\ln P(n)$ vs. $\ln n$ for $\epsilon = \{0 \text{ (solid blue line)}, 0.05 \text{ (dashed red line)}\}$, $\tilde{\sigma}_e = 0.01$. The distribution for $\epsilon = 0$ is typical of the egalitarian phase with a large slope for large $n$. The distribution for $\epsilon = 0.05$, however, has the slower falloff characteristic of the dominance phase. In addition, there is a shoulder for very large $n$, which is due to the existence of multiple sets of several transient quasi-stable species, arising from the chaotic dynamics present in the absence of environmental stochasticity.

(*Kessler and Shnerb, 2025*). Such a clique gives rise to a 'shoulder' at high abundance values (see *Figure 5*), since at any given moment there are several species with high abundance. In contrast, the hierarchical structure under pure environmental stochasticity leads to a simple power law, as seen earlier in *Figure 1*.

As $\epsilon$ increases from zero to a finite value, the neutral-stochastic system discussed throughout this article transforms into a system where the dominant factor is deterministic chaos. In the chaotic phase, the dynamics still appears stochastic, but not due to fluctuations in external parameters, but rather as a result of the inherent characteristics of the deterministic dynamics.

*Figure 4* demonstrates that the transition to chaos affects $S_{1/2}$ in opposite ways in the two phases. In the egalitarian phase, the main effect of chaos is to increase the effective stochasticity, leading to a decline of $S_{1/2}$. In the dominance phase, by contrast, $S_{1/2}$ increases. This is due to the 'shoulder' in the abundance distribution described in *Figure 5*, that is, because dominance in the chaotic regime is associated with a clique rather than a single species.

## Discussion

Hyperdiverse communities, like those analyzed in this article, are extremely important and frequently occur in nature. However, quantifying their specific parameters is an impossible task. Therefore, the attempt to understand the factors that dictate the community structure in these systems requires the use of models, which should preferably be as generic as possible.

Broadly speaking, there are three generic scenarios for the dynamics of multispecies systems: those involving a stable clique of resident species; those in which the composition of the group of high-abundance species changes over time due to chaotic dynamics; and those assuming neutral dynamics.

In the first case, the set of high-abundance species is determined by their relative fitness (which reflects their intrinsic growth rate and interspecific interactions), which remains fixed over time. In the second, the composition of the dominant cliques varies intermittently over time, but this turnover is governed by deterministic factors, and at any given moment, only a specific set of low-abundance species is capable of exponential growth. Moreover, the system admits a characteristic timescale, the time required for one of these rare species to invade. This timescale depends logarithmically on the migration rate $\mu$.

In the third case, discussed here, the identity of the high-abundance species at any given moment is a random outcome of environmental stochasticity. As in the chaotic case, there is no sharp separation between core species and low-abundance ones, and the fundamental timescale of the system is determined by the stickiness phenomenon described above.

As far as can be judged from the empirical SADs of *Figure 2*, it appears that they cannot be explained by the first scenario. Two of its main features are a gap between species with abundance of $\mathcal{O}(K)$ and those with abundance of $\mathcal{O}(\mu)$, and a truncated Gaussian-shaped SAD for the high-abundance species – both of which are not observed in our distributions.

The chaotic case yields SADs that more closely resemble those we have reviewed here: they exhibit no gap and display a power-law behavior. In fact, when attempting to derive an effective equation for a single species, one arrives, as noted above, at the very same equation. As is well known, it is often possible to identify multiple underlying models that give rise to the same abundance distribution. Therefore, matching the SAD is a necessary condition for considering a model a plausible candidate, but leaves open the possibility of alternative explanations.

To tell apart chaos from environmental stochasticity, it may be necessary to focus specifically on the dynamics of the most abundant species. If the noise is intrinsic, originating from chaotic fluctuations, then a species that has reached dominance is expected to exhibit fluctuation patterns that differ markedly from those observed when the same species is rare. In contrast, if the fluctuations stem from external factors such as weather or precipitation, we would expect the statistical properties of abundance fluctuations to be independent of the species' current abundance level, as has been observed empirically (*Kalyuzhny et al., 2014b*).

Another test that may help distinguish between a chaotic and a stochastic system relates to the emergence of a fundamental timescale (*Arnoulx de Pirey and Bunin, 2024*) in chaotic dynamics – one associated with the invasion of rare species. Such a timescale does not appear in systems driven by external stochasticity. Like the previous approach, this too requires one to examine dynamic properties of the system, rather than relying solely on static features such as the abundance distribution.

Whatever the underlying mechanism, the net result, as we have demonstrated herein, is that in the limit of strong interspecies coupling, a transition occurs between an egalitarian phase and a dominance phase. When the immigration rate $\mu$ is reduced, total carrying capacity $K$ increases, or when environmental stochasticity $\sigma_e$ increases, the system can suddenly lose a significant amount of diversity at the transition point. The dependence on the total population size $K$ is particularly interesting and has to do with the dispute about the relationships between productivity and species richness (*Waide et al., 1999*; *Kadmon and Benjamini, 2006*).

We will now discuss some potential extensions of the model described here and explore their possible implications.

Our model assumed an uncorrelated response of species to environmental variations. This treatment can be easily extended to include *correlated responses*, using techniques similar to those implemented by *Loreau and de Mazancourt, 2008*. In general, under correlated responses, the effective number of species in the community decreases.

Another interesting point relates to the interplay between the stickiness mechanism, through which environmental stochasticity causes populations to spend long periods in a state of rarity, and mechanisms such as the *storage effect or relative nonlinearity* (*Chesson, 1982*; *Chesson, 2000*; *Ellner et al., 2016*; *Letten et al., 2018*) that allow rare populations to invade due to environmental stochasticity. It is likely that these mechanisms weaken as the number of species increases (for storage, this has been demonstrated in several studies; *Chesson and Huntly, 1989*; *Pande and Shnerb, 2022*), and therefore, at least in diverse communities of competing species, the dominant effect will actually be that of stickiness.

The time-averaged neutral dynamic considered here assumes that the differences in average fitness between species are negligible, at least to a first approximation. This assumption is necessary in situations where niche overlap is large; otherwise, fitness differences would cause the extinction of most species. The justification for this can come from processes leading to emergent neutrality (*Holt, 2006*; *Vergnon et al., 2012*), or from the fact that environmental stochasticity itself is also a mechanism that 'neutralizes' fitness differences (*Pande and Shnerb, 2022*). Extending our model as we have done above to include *weak deviations* from time-averaged neutrality, perhaps using the dynamic

mean-field approximation (*Roy et al., 2019*) , is a critical first step, which remains to be explored in more detail.

Species coexistence has long been, and remains, a theoretical puzzle of immense importance for understanding the dynamics of biological systems, with far-reaching practical implications. The research conducted in recent years has provided powerful theoretical tools that allow us to focus the discussion and understand the generic implications of community structure and the nature of species interactions on the range of possible outcomes. We believe that the parameter range we have addressed in this article – (nearly-) neutral dynamics and significant environmental stochasticity – is relevant for a wide variety of ecological systems, and we hope that our work will serve as a foundation for further studies that explain the wide range of diversity levels (for example, between a tropical forest and a tundra, or between different types of microbiome) in relation to the phase transition described here.

## Materials and methods

### Data compilation and analysis

As explained below, the phase transition we analyze in this paper manifests itself in two characteristics: the SAD and the egality parameter $S_{1/2}/S$. Like any phase transition in complex systems, the distinction between the two phases becomes sharper as the system grows larger, so that $S$ increases. To test our predictions, we used several recent large databases on communities with thousands of species: human microbiome (*David et al., 2014*), marine prokaryotes (*Eguíluz et al., 2019*), tropical forest trees (*Cooper et al., 2024*), and birds (worldwide) (*Callaghan et al., 2021*).

### The (noise-free) neutral model

We consider a community of $S$ populations, with differential responses to environmental variations. $N_i$ is the abundance of the $i$th species, and the carrying capacity parameter is denoted by $K$. With no stochasticity and no immigration, $N_i$ satisfies the equation

$$\frac{dN_i}{dt} = N_i \left( 1 - \frac{\sum_{j=1}^{S} N_j}{K} \right). \tag{8}$$

As all species admit the same growth rate and interact in a symmetric manner, the model is neutral. In particular, any solution that satisfies $\sum_{j=1}^{S} N_j = K$ is a steady state of the system.

### The time-averaged neutral model

To allow for nontrivial dynamics, one would like to add environmental stochasticity and immigration to the process described in *Equation 8*. The rate of immigration is denoted by $\mu$, and the growth rate of each species fluctuates in time, undergoing an Ornstein–Uhlenbeck process with a correlation time $\tau$. The corresponding set of stochastic differential equations, as set out in *Equation 1*.

$$\frac{dN_i}{dt} = \mu + N_i \left( 1 - \frac{\sum_{j=1}^{S} N_j}{K} \right) + r_i(t)N_i$$

$$\frac{dr_i}{dt} = -\frac{r_i}{\tau} + \theta\eta_i(t), \tag{9}$$

where $\eta_i(t)$ is a white-noise process of unit strength. The index $i$ of $\eta_i$ indicates that each of the $S$ species responds to the environmental variations in an independent manner. The correlation function of $r_i$ is

$$\langle r_i(t)r_j(t') \rangle = \delta_{i,j} \frac{\theta^2 \tau}{2} e^{-|t-t'|/\tau}. \tag{10}$$

The equivalent white-noise strength of the environmental stochasticity, twice the diffusion constant of $\log N_i$, is then given by $\tilde{\sigma}_e^2 = \theta^2 \tau^2$, the time integral of the $r_i$ correlation function.

### Fokker–Planck (FP) equations

The main analytical results presented in this article are based on solving FP equations. To derive these equations, we assumed a relatively short correlation time $\tau$ for the environmental stochasticity, which allows the use of a single effective equation for the process described in *Equation 8*. The relevant considerations are detailed in 'SI B'.

### Demographic stochasticity

This refers to the intrinsic randomness of the birth–death process at the individual level. This stochasticity also manifests in erratic abundance variations, but the intensity of these fluctuations is weaker (compared to environmental stochasticity) when the population is large, and therefore it was not included in *Equation 8* or in the analyses of *van Nes et al., 2024* and *Mallmin et al., 2024*. However, demographic stochasticity is crucially important in small populations and in extinction processes; as we have seen, it must be considered outside the microorganism realm. To include demographic stochasticity in the treatment, one must add a term to the relevant equation that expresses noise whose amplitude scales with the square root of the population size, as opposed to environmental stochasticity, whose amplitude scales linearly with population size. The technical details can be found in 'SI D.

## Acknowledgements

We thank Emil Mallmin and Egbert van Nes for the interesting exchange of ideas that contributed to the shaping of this work. NMS acknowledges funding from the Israeli Ministry of Innovation, Science and Technology (MOST), under the framework of the Israel–Italy Cooperation Program. DAK acknowledges funding from the Israel Science Foundation, Grant 1614/21.

## Additional information

### Funding

| Funder | Grant reference number | Author |
| --- | --- | --- |
| Israeli Ministry of Innovation, Science and Technology | Israel-Italy Cooperation Program | Nadav M Shnerb |
| Israel Science Foundation | 1614/21 | Nadav M Shnerb |

The funders had no role in study design, data collection and interpretation, or the decision to submit the work for publication.

### Author contributions

David A Kessler, Nadav M Shnerb, Conceptualization, Formal analysis, Investigation, Methodology, Writing – original draft, Writing – review and editing

### Author ORCIDs

David A Kessler ⓘ https://orcid.org/0000-0002-5279-1655
Nadav M Shnerb ⓘ https://orcid.org/0000-0003-4418-6284

### Decision letter and Author response

Decision letter https://doi.org/10.7554/eLife.103999.sa1
Author response https://doi.org/10.7554/eLife.103999.sa2

## Additional files

### Supplementary files

MDAR checklist

## Data availability

The current manuscript is a computational study, so no data have been generated for this manuscript. The simulation code can be accessed at https://github.com/kesslerda11/Sticky (copy archived at *Kessler, 2024*). The data analyzed was obtained from presvious published works, as indicated in the References.

The following previously published datasets were used:

| Author(s) | Year | Dataset title | Dataset URL | Database and Identifier |
|---|---|---|---|---|
| David LA, Materna AC, Friedman J, Campos-Baptista MI, Blackburn MC, Perrotta A, Erdman SE, Alm EJ | 2014 | Host lifestyle affects human microbiota on daily timescales - Subject A gut counts | https://www.ebi.ac.uk/ena/browser/view/PRJEB6518 | European Nucleotide Archive, PRJEB6518 |
| Eguíluz VM, Salazar G, Fernández-Gracia J, Pearman JK, Gasol JM, Acinas SG, Sunagawa S, Irigoien X, Duarte CM | 2015 | MolEcol_2015 | https://github.com/GuillemSalazar/MolEcol_2015/blob/master/OTUtable_Salazar_etal_2015_Molecol.txt | GitHub, OTUtable_Salazar_etal_2015_Molecol.txt |
| Cooper D, Lewis S, ter Steege H, Sullivan M, Slik JWF, Barbier N | 2024 | Cooper consistent patterns of common species across tropical forest tree communities | https://doi.org/10.6084/m9.figshare.21670883.v1 | figshare, 10.6084/m9.figshare.21670883.v1 |

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

# Appendix 1

## Supporting information

### SI A: The necessity of immigration

*van Nes et al., 2024* have considered various models that were proposed for a diverse community in which all species have the same mean fitness. Most cases were reviewed in the supplemental material of the aforementioned article, but the version analyzed in detail in the main text is

$$\frac{dN_i}{dt} = N_i \left( 1 - \frac{\sum_{j=1}^{S} N_j}{K} \right) dt + \tilde{\sigma}_e \eta_i(t) N_i, \tag{SA1}$$

where $\eta(t)$ is a white noise process. Ito calculus is applied, so the numerical integration procedure satisfies

$$N_{i,t+\Delta t} = N_{i,t} + \Delta t \left[ \mu + N_{i,t} \left( 1 - \frac{\sum_{j=1}^{S} N_{j,t}}{K} \right) \right] + \tilde{\sigma}_e \sqrt{\Delta t} \mathcal{N}(0,1) N_{i,t}, \tag{SA2}$$

where $\mathcal{N}(0,1)$ is a random variable drawn (independently for each $\Delta t$ and each species) from a normal distribution with zero mean and unit variance.

However, in the absence of an immigration term, that is, when $\mu = 0$, such a process leads to monodominance, the state where only one species survives and the abundance of all other species drops below any finite value. *Figure 1—figure supplement 1* demonstrates numerically this behavior. The results presented in our main text explain this phenomenon. First, if $\mu = 0$ then $\alpha = 0$ in our solution, *Equation 4*. The resulting species abundance distribution diverges at zero abundance and cannot be normalized. The interpretation of such non-normalizable abundance distribution is that the abundance of all species will drop below any finite threshold as time goes to infinity.

One may arrive at the same conclusion from a different perspective (*Dean and Shnerb, 2020*). Let's assume we have two species in a community with a fixed and rigid carrying capacity (i.e., they are playing a zero-sum game) and environmental stochasticity that determines their relative fitness.

Denoting by $x$ the fraction of the focal species, its dynamics satisfies $dx/dt = s(t)x(1-x)$, where $s(t)$ reflects the effect of the fluctuating environment. Therefore, the population undergoes a balanced walk in the logit space $z = \ln[x/(1-x)]$. In this type of random walk, a species that 'gets stuck' in the low-density region will remain there for increasing periods, as described in Section 5 of *Dean and Shnerb, 2020*. This is the 'diffusive trapping', or the stickiness of low-abundance states, that causes the system to be dominated at any given moment by a single species.

Now we can easily generalize this argument for $S$ species: the relative fitness of each species (i.e., its fitness relative to the average fitness of the community) is a zero-mean random variable. Therefore, in the log-abundance space, it undergoes a random walk, and so all species except one will fall below any threshold of abundance over time. Thus, as seen in *Figure 1—figure supplement 1*, the system in the long run is dominated by a single species, and the abundance of all other species is negligible.

### SI B: Derivation of *Equation 4*

The FP equation that corresponds to *Equation 2* is

$$\frac{\partial P(n,t)}{\partial n} = \frac{\sigma^2}{2} \frac{\partial^2}{\partial n^2} \left( n^2 P(n) \right) + \left( c - \frac{\sigma^2}{2} \right) \frac{\partial}{\partial n} \left( n P(n) \right) - \mu \frac{\partial P(n)}{\partial n} \tag{SB1}$$

In steady state, $P(n)$ is time-independent and so its time derivative vanishes, leaving one to solve the exact ordinary differential equation (after first integration),

$$\frac{\sigma^2}{2} \frac{\partial}{\partial n} \left( n^2 P(n) \right) + \left( c - \frac{\sigma^2}{2} \right) \left( n P(n) \right) - \mu P(n) = 0. \tag{SB2}$$

Here, we have put the integration constant to zero to avoid diverging solutions that yield non-normalizable $P(n)$ functions. Therefore,

$$\frac{\sigma^2}{2}n^2\frac{\partial P(n)}{\partial n} + \left(c + \frac{\sigma^2}{2}\right)nP - \mu P(n) = 0. \tag{SB3}$$

That yields **Equation 4** of the main text.

## SI C: From egalitarian community to hyperdominant species

Let us consider a community of $J$ individuals, whose species richness is $S$. By definition,

$$S = \int_0^\infty P(n)dn, \tag{SC1}$$

and

$$J = \int_0^\infty nP(n)dn, \tag{SC2}$$

From **Equation 4** of the main text, we take $P(n)$ and plug it into **Equation SC1**. First, we implement the condition on $S$ to determine the normalization factor $A$, and find

$$P(n) = S\frac{\alpha^{\beta-1}}{\Gamma[\beta-1]}e^{-\alpha/n}n^{-\beta} \tag{SC3}$$

Now let us assume $\beta > 2$. In that case, the integral in (S2) converges so that

$$J = \frac{\alpha}{\beta-2}S. \tag{SC4}$$

As a metric for the egality of the community, let us take the one used in **van Nes et al., 2024**. First, we will check the richness $S_{1/2}$ of the minimum set of species required to contain at least 50% of the entire community population. If the species within this set with the smallest abundance is represented by $n_{1/2}$ individuals, this implies (implementing **Equation SC4**).

$$\int_{n_{1/2}}^\infty nP(n)dn = \alpha S\left((\beta-1) - \frac{\Gamma[\beta-2,\alpha/n_{1/2}]}{\Gamma[\beta-1]}\right) = J/2 = \frac{\alpha}{2(\beta-2)}S, \tag{SC5}$$

When $\mu \to 0$ then $\alpha \to 0$ and hence,

$$\Gamma[\beta-2,\alpha/n_{1/2}] \approx \Gamma[\beta-2] - \frac{1}{\beta-2}\left(\frac{\alpha}{n_{1/2}}\right)^{\beta-2} \tag{SC6}$$

therefore

$$\frac{\alpha}{n_{1/2}} = \left(\frac{\Gamma[\beta-1]}{2}\right)^{1/(\beta-2)}, \tag{SC7}$$

so the ratio $\alpha/n_{1/2}$ approaches zero as $\beta \to 2^+$

The fraction of species whose abundance is equal or greater than $n_{1/2}$ is obtained as

$$\frac{S_{1/2}}{S} = \frac{1}{S}\int_{n_{1/2}}^\infty P(n)dn = 1 - \frac{\Gamma[\beta-1,\alpha/n_{1/2}]}{\Gamma[\beta-1]} = \frac{1}{\Gamma[\beta]}\left(\frac{\Gamma[\beta-1]}{2}\right)^{\frac{\beta-1}{\beta-2}}. \tag{SC8}$$

In particular, as $\beta \to 2^+$,

$$\frac{S_{1/2}}{S} \sim e^{-(\ln 2)/(\beta-2)}. \tag{SC9}$$

## SI D: Demographic stochasticity

Every group of living organisms is influenced by various types of stochasticity. The literature (*Kalyuzhny et al., 2014b*; *Lande and Engen, 2003*) usually distinguishes between *environmental stochasticity*, which affects populations coherently, and *demographic stochasticity*, which affects the fitness of individuals or small groups of individuals in an incoherent manner, so that the overall reproductive success of the population reflects the sum of many random events. Weather (such as temperature or precipitation) fluctuations that are correlated over a large area and simultaneously affect entire populations contribute to environmental stochasticity, while the degree of success or failure of individuals in finding food or avoiding predators relative to its population mean contributes to demographic stochasticity.

Environmental stochasticity is represented in equations by noise that is proportional to the population size. Thus, in *Equation 1* of the main text, the term $r_i(t)$, which varies stochastically, multiplies $N_i$. This mathematical expression reflects the situation where, in a good year, the average number of offspring for each individual increases, and in a bad year, it decreases. Demographic stochasticity, on the other hand, represents the success or failure of individuals, so that the overall result, according to the Central Limit Theorem, is fluctuations that are proportional to the square root of the population size. The corresponding term in the Langevin equation is $\eta(t)\sqrt{N_i}$.

In an FP equation, the term that represents demographic stochasticity is

$$\frac{\sigma_d^2}{2} \frac{\partial^2}{\partial n^2} \left( nP(n) \right) \tag{SD1}$$

We denote the strength of demographic stochasticity (i.e., the variance in the number of offspring per individual) by $\sigma_d^2$, distinguishing it from $\sigma_e^2$, which represents the strength of environmental stochasticity.

The extended version of *Equation SD1* thus takes the form

$$\frac{\partial P(n,t)}{\partial n} = \frac{\sigma_e^2}{2} \frac{\partial^2}{\partial n^2} \left( n^2 P(n) \right) + \frac{\sigma_d^2}{2} \frac{\partial^2}{\partial n^2} \left( nP(n) \right) + \left( c - \frac{\sigma^2}{2} \right) \frac{\partial}{\partial n} \left( nP(n) \right) - \mu \frac{\partial P(n)}{\partial n} \tag{SD2}$$

Again, in steady state, $P(n)$ is time-independent,

$$\frac{\sigma_e^2}{2} \frac{\partial}{\partial n} \left( n^2 P(n) \right) + \frac{\sigma_d^2}{2} \frac{\partial}{\partial n} \left( nP(n) \right) + \left( c - \frac{\sigma_e^2}{2} \right) \left( nP(n) \right) - \mu P(n) = 0, \tag{SD3}$$

and therefore the species abundance distribution is given by

$$P(n) = An^{-1+2\mu/\sigma_d^2} \left( \frac{\sigma_d^2}{\sigma^2} + n \right)^{-2c/\sigma^2 - 2\mu/\sigma_d^2} \tag{SD4}$$

## SI E: *Figure 3* information

For each dataset, the data was log-binned (scaled by the inverse of the width of the bin), then the log-log histogram was fitted with the relevant expression.

**Appendix 1—table 1.** Summary of fitted models and parameter estimates (with 1σ uncertainties) for the four systems analyzed.

| System (panel) | Fitted equation | Fitted parameters (±1σ) |
|---|---|---|
| Gut microbiome (a) | Fit to *Equation 4* with $\alpha = 2\mu/\sigma_e^2$, $\beta = 1 + 2c/\sigma_e^2$, A normalization factor | $A = 1.77 \pm 0.060$; $\alpha = 5.5 \times 10^{-6} \pm 5 \times 10^{-7}$; $\beta = 1.31 \pm 0.01$ |

*Appendix 1—table 1 Continued on next page*

*Appendix 1—table 1 Continued*

| System (panel) | Fitted equation | Fitted parameters (±1σ) |
|---|---|---|
| Plankton (b) | Fit to **Equation 4** with<br>$\alpha = 2\mu/\sigma_e^2$,<br>$\beta = 1 + 2c/\sigma_e^2$,<br>$A$ normalization factor | $A = 6.7 \pm 0.2$;<br>$\alpha = 0.8 \pm 0.5$;<br>$\beta = 1.7 \pm 0.03$ |
| Birds (c) | Fit to **Equation 6** with<br>$\gamma = 2\mu/\sigma_d^2$,<br>$\beta = 1 + 2c/\sigma_e^2$,<br>$k = \sigma_d^2/\sigma_e^2$,<br>$A$ normalization factor | $A = 23 \pm 1$;<br>$\gamma = 0.34 \pm 0.01$;<br>$\beta = 2.05 \pm 0.05$;<br>$k = 3.0 \times 10^6 \pm 6 \times 10^5$ |
| Amazon trees (d) | $A = 14 \pm 1$;<br>$\gamma = 2\mu/\sigma_d^2$,<br>$\beta = 1 + 2c/\sigma_e^2$,<br>$k = \sigma_d^2/\sigma_e^2$,<br>$A$ normalization factor | $A = 14 \pm 1$;<br>$\gamma = 0.013 \pm 0.096$;<br>$\beta = 2.4 \pm 0.2$;<br>$k = 380 \pm 200$ |

## SI F: Stochastic and chaotic dynamics

This figure shows the abundance of 200 species over time, in the purely stochastic case ($\epsilon = 0$) and in the case dominated by chaotic dynamics ($\epsilon = 0.05$). We observe that chaotic dynamics introduce a new element into the system: a clique of species that are "dominant" at any given moment. Although there is always a species that can invade this clique and drive one or more of its members to low abundance levels, this process takes time. As a result, the number of highly abundant species is larger compared to the stochastic case, where we see a strong dominance of a single species at each moment.

