## [Editor Report]

This valuable paper explores the question of when complex ecosystems will be dominated by a few species. The authors present compelling, general arguments for a phase transition from what they call a dominance phase (few species dominate biomass) to an egalitarian phase (no species dominates the biomass).

---

## [Decision Letter]

**Decision letter after peer review:**

Thank you for submitting your article "Dominance to egalitarian transition in diverse communities" for consideration by *eLife*. Your article has been reviewed by 2 peer reviewers, and the evaluation has been overseen by a Reviewing Editor and Sergio Rasmann as the Senior Editor.

Essential Revisions:

As you can see below, while both reviewers appreciated the work and its contribution to the field, they raise several technical comments as well as ones pertaining to the presentation. The work will be strengthened if these are addressed in a revision.

*Reviewer #1 (Recommendations for the authors):*

In this manuscript, the authors investigate the effects of environmental noise on the dynamics of species in hyper-diverse communities. They assume that the community is "time-averaged neutral", namely follows a neutral model (inter- and intra-species interactions are the same), but subject to different and statistically-independent environmental noises. They calculate the species abundance distribution (SAD) and find that the highly-abundant (dominant) species consist of either a finite fraction of the species, or only a small subset of species (growing sub-linearly with the total number of species). The results are compared with SADs from field data.

The manuscript makes two additional contributions:

(1) The discussion leading to these results takes the authors to recent popular concepts in the field such as "diffusive trapping" and "stickiness". In this context, they point out a problem with a recent paper that has received much attention.

(2) They incorporate demographic noise, which I personally find to be the most original part of the work. Comparing the SADs with data, they argue that this term is important even in situations where one would expect a large number of individuals of each species.

The manuscript makes important contributions that merit publication in *eLife*, is well-structured and overall well-written. However, I found that there are issues in its present form that could be improved. This includes certain aspects of the presentation, and the comparison with other phenomena, in particular abundance fluctuations solely due to the interactions between species, to which the authors dedicate much of the Discussion section.

My particular comments are as follows:

1. Regarding the presentation. I felt that Sec. IIIC that applies the theory to data, is written in a confusing way. It says: "Figure 3 shows that the formula we presented … Equation (4) … describes the species abundance distribution for the gut microbiome quite well." But what was actually done with the data is not clear. I assume that Equation. (4) was fit to the data. But then the discussion goes on to compare with the result with demographic noise, which is given only later in the text (Equation. (6)). All this, without clearly pointing out which panel in Figure 3 is compared with which expression, and what parameters are being fit. Also, it would be helpful to include a table with the values of all parameter values obtained by the fit.

2. In this context, it should be mentioned that the ability to fit the SADs to the expressions is certainly an important first step, but it is not a strongly selective criterion between theories. It is well-known that many theories can fit the same SADs. I feel that this point should at least be mentioned in the text (perhaps in the Discussion).

3. A large part of the Discussion is dedicated to comparison of the model in question with chaotic dynamics due to the interactions between the species. There are several issues with this Discussion:

(a) In line 217, right below Equation. (6), the authors make a distinction between \begin{document}$\alpha_{ij}$\end{document} that are "O(1) in the strong coupling scenario, and O(1/S) in the weak coupling scenario". This distinction is unclear and seems arbitrary. First of all, it is not clear what is meant by these scalings; one has to explain how the distribution of the values P(\begin{document}$\alpha_{ij}$\end{document}) changes with S, and at least distinguish between the changes to the mean and the standard deviation of the distribution. Secondly, these two scalings (whatever moments they refer to) seem arbitrary; the transition to chaos happens when the distribution width \begin{document}$\text{std}(\alpha)\sim1/S^{1/2}$\end{document} which is neither of the above mentioned. Finally, these terms are not used in a consistent way, and the term "high niche overlap" is used later (referring to the strong coupling, perhaps?)

(b) Line 219 says that "weakly interacting species" lead to a truncated Gaussian distribution. The truncated Gaussian is only obtained at a fixed point phase (and only for interaction distributions that have a finite variance). In the chaotic phase, for any scaling which allows for chaos, wide distributions are obtained.

(c) Regarding sentence starting on line 220: "Remarkably, even in the strongly interacting case, … , each species can be treated separately …". This has been shown numerically to give reasonable results, but I'm not sure that it is known to be an exact dynamical description in the "strong interacting case" as the authors define it.

(d) Line 224: "linear growth rate is weak and negative." This is true for any dynamics that yield chaos, as long as the fluctuation of the interaction sum-term in Equation. (6), are comparable in size to the other terms in the parenthesis in that equation.

(e) Line 243: "After all, it is well known that distinguishing between high-dimensional chaos and mere noise is virtually impossible." While this is true for standard chaos, it is not the case here, exactly because of the phenomena relating to the "heteroclinic" structure of the equations, that leads to phenomena such as "stickiness" that the authors refer to earlier. Some fingerprints are: (1) the internal timescale of the chaotic dynamics that need not at all be related to the timescales of environmental driving, and may be large for small migration. (2) SADs of chaotic dynamics, as a result of the previous point. (3) Changing the number of species one might cross the transition to chaos, and either see stable equilibria or fluctuations, in contrast with changes due to environmental changes.

4. Finally, let me go back and say that I liked the inclusion of demographic noise in the model, and it is interesting that it plays a role in some of the data sets, even when population sizes would naively appear to be quite large (well above tens of individuals, where effects of demographic noise are usually important). If possible, I would have liked to see some discussion of this point. When is demographic noise expected to matter in this model, and what does it mean for the communities in question?

*Reviewer #2 (Recommendations for the authors):*

In their paper "Dominance to egalitarian transition in diverse communities", Kessler and Shnerb explore the question of if/when diverse ecosystems will be in a dominance phase (few species dominate biomass) or an egalitarian phase (no species dominates the biomass). To do this, the authors build on their previous work exploring time-averaged neutral models. The major technical advance of the work is to carefully use the Stratonovich formalism to show that, even though on average the growth rate is zero, given noise one will get an extra fluctuation-dependent contribution to the growth rate. This careful treatment shows that in this class of models, there exists a phase transition between these phases depending on noise and immigration rates.

The technical calculations are straight forward and seem to correct (these are mostly in the appendix). The major finding of the paper is a transition from what they call an egalitarian regime (\β>2 in Equation 4) to a dominant regime (\β <2) in the SAD distribution. The authors then fit various SAD curves for four experimental systems in Figure 3.

Overall, I found the paper quite compelling and thoughtful. I thought the problem was well motivated and the arguments were well made.

I do have some questions and comments that I think could make the paper a little stronger.

1. I would like much more discussion and motivation about the difference between \begin{document}$\tilde{\sigma_e}$\end{document} (defined on top of page 3 below Equation 2) and the phenomenological parameter \σ_e in Equation 3. I am not sure what is exactly being assumed here and why the different notation?\begin{document}$$\displaystyle {\sigma_e}$$\end{document}

2. The authors use a very particular definition of egalitarian and dominance based on the number of species needed to explain half the biomass and how it scales with the size of the ecosystem S? As a statistical physicist, I am wondering if alternative definitions of these quantities such as the Inverse participation Ratio (IPR) also show similar scaling?

3. The authors make a cryptic statement about seeing only the tails of the ocean prokaryotic data. I would like much more discussion of this.

4. Also, given the very close values of the empirical fits of \β to the critical \β_c=2, it would be very useful to have some sense of error bars? For example, is \β=2.03 really over the transition? I would like some bootstrapped error bars for these fit parameters.

5. It would be nice to see some simulations for systems that are not strictly neutral and how well these can be approximated by the time-dependent neutral model. How large does the LV interaction matrix (A in Equation. 7) have to be before the arguments here break down for a simulated community?

I think my primary substantial concern is about error bars on the fits. This is quite important for drawing conclusions about the real communities.

---

## [Author Response]

Essential Revisions:As you can see below, while both reviewers appreciated the work and its contribution to the field, they raise several technical comments as well as ones pertaining to the presentation. The work will be strengthened if these are addressed in a revision.Reviewer #1 (Recommendations for the authors):In this manuscript, the authors investigate the effects of environmental noise on the dynamics of species in hyper-diverse communities. They assume that the community is "time-averaged neutral", namely follows a neutral model (inter- and intra-species interactions are the same), but subject to different and statistically-independent environmental noises. They calculate the species abundance distribution (SAD) and find that the highly-abundant (dominant) species consist of either a finite fraction of the species, or only a small subset of species (growing sub-linearly with the total number of species). The results are compared with SADs from field data.The manuscript makes two additional contributions:(1) The discussion leading to these results takes the authors to recent popular concepts in the field such as "diffusive trapping" and "stickiness". In this context, they point out a problem with a recent paper that has received much attention.(2) They incorporate demographic noise, which I personally find to be the most original part of the work. Comparing the SADs with data, they argue that this term is important even in situations where one would expect a large number of individuals of each species.The manuscript makes important contributions that merit publication in eLife, is well-structured and overall well-written. However, I found that there are issues in its present form that could be improved. This includes certain aspects of the presentation, and the comparison with other phenomena, in particular abundance fluctuations solely due to the interactions between species, to which the authors dedicate much of the Discussion section.

We thank Reviewer #1 for the positive and constructive review, and for highlighting the key contributions of our work

My particular comments are as follows:1. Regarding the presentation. I felt that Sec. IIIC that applies the theory to data, is written in a confusing way. It says: "Figure 3 shows that the formula we presented … Equation (4) … describes the species abundance distribution for the gut microbiome quite well." But what was actually done with the data is not clear. I assume that Equation. (4) was fit to the data. But then the discussion goes on to compare with the result with demographic noise, which is given only later in the text (Equation. (6)). All this, without clearly pointing out which panel in Figure 3 is compared with which expression, and what parameters are being fit. Also, it would be helpful to include a table with the values of all parameter values obtained by the fit.

We have rewritten sec IIIc in a way that avoids confusion between the discussion of Eq 6 (and the corresponding panels a and b in figure 3) and the discussion of demographic stochasticity, Eq 7 and panels c and d of Figure 3. We also added the values of the fitting parameters in a detailed table (Table 1) presented in the SI.

2. In this context, it should be mentioned that the ability to fit the SADs to the expressions is certainly an important first step, but it is not a strongly selective criterion between theories. It is well-known that many theories can fit the same SADs. I feel that this point should at least be mentioned in the text (perhaps in the Discussion).

We do agree that SADs are not uniquely determinative of the underlying dynamics, and that additional dynamical information is crucial to make a more definitive statement. In the revised version we have added a statement along these lines in the Discussion section.

Besides, our goal in this paper is to point out the qualitatively different SADs in different empirical systems and to suggest a simple theory that accounts for these in the general context of nearly-neutral theories. We agree that identifying the precise mechanism requires more dynamic data; our focus here is to propose a simple theoretical framework that captures the observed SAD patterns under the assumption of near-neutrality.

3. A large part of the Discussion is dedicated to comparison of the model in question with chaotic dynamics due to the interactions between the species. There are several issues with this Discussion:(a) In line 217, right below Equation. (6), the authors make a distinction between \begin{document}$\alpha_{ij}$\end{document} that are "O(1) in the strong coupling scenario, and O(1/S) in the weak coupling scenario". This distinction is unclear and seems arbitrary. First of all, it is not clear what is meant by these scalings; one has to explain how the distribution of the values \begin{document}$P(\alpha_{ij})$\end{document} changes with S, and at least distinguish between the changes to the mean and the standard deviation of the distribution. Secondly, these two scalings (whatever moments they refer to) seem arbitrary; the transition to chaos happens when the distribution width \begin{document}$\text{std}(\alpha)\sim1/S^{1/2}$\end{document} which is neither of the above mentioned. Finally, these terms are not used in a consistent way, and the term "high niche overlap" is used later (referring to the strong coupling, perhaps?)(b) Line 219 says that "weakly interacting species" lead to a truncated Gaussian distribution. The truncated Gaussian is only obtained at a fixed point phase (and only for interaction distributions that have a finite variance). In the chaotic phase, for any scaling which allows for chaos, wide distributions are obtained.

We thank the reviewer for this illuminating comment. We do agree that the terminology employed in this part of the Discussion section was unclear. In the revised manuscript we have completely reformulated our presentation, which is now focused on the three options for explaining the features of diverse competing communities: stable, chaotic and nearly neutral.

(c) Regarding sentence starting on line 220: "Remarkably, even in the strongly interacting case, … , each species can be treated separately …". This has been shown numerically to give reasonable results, but I'm not sure that it is known to be an exact dynamical description in the "strong interacting case" as the authors define it.

The sentence quoted here does not appear in the revised version. We do refer elsewhere to the fact that the analysis of chaotic dynamics made use of the same single-species equation, but of course we do not claim that this reduction is exact or rigorous.

(d) Line 224: "linear growth rate is weak and negative." This is true for any dynamics that yield chaos, as long as the fluctuation of the interaction sum-term in Equation. (6), are comparable in size to the other terms in the parenthesis in that equation.

This sentence no longer appears in the revised discussion.

(e) Line 243: "After all, it is well known that distinguishing between high-dimensional chaos and mere noise is virtually impossible." While this is true for standard chaos, it is not the case here, exactly because of the phenomena relating to the "heteroclinic" structure of the equations, that leads to phenomena such as "stickiness" that the authors refer to earlier. Some fingerprints are: (1) the internal timescale of the chaotic dynamics that need not at all be related to the timescales of environmental driving, and may be large for small migration. (2) SADs of chaotic dynamics, as a result of the previous point. (3) Changing the number of species one might cross the transition to chaos, and either see stable equilibria or fluctuations, in contrast with changes due to environmental changes.

Again, this specific sentence no longer appears in the revised discussion, which was rewritten in line with the comments of the reviewer.

4. Finally, let me go back and say that I liked the inclusion of demographic noise in the model, and it is interesting that it plays a role in some of the data sets, even when population sizes would naively appear to be quite large (well above tens of individuals, where effects of demographic noise are usually important). If possible, I would have liked to see some discussion of this point. When is demographic noise expected to matter in this model, and what does it mean for the communities in question?

We have revised the discussion around Eq. (8) to clarify the role of demographic stochasticity and the reasons for its pronounced effect even for relatively large abundance values.

Reviewer #2 (Recommendations for the authors):In their paper "Dominance to egalitarian transition in diverse communities", Kessler and Shnerb explore the question of if/when diverse ecosystems will be in a dominance phase (few species dominate biomass) or an egalitarian phase (no species dominates the biomass). To do this, the authors build on their previous work exploring time-averaged neutral models. The major technical advance of the work is to carefully use the Stratonovich formalism to show that, even though on average the growth rate is zero, given noise one will get an extra fluctuation-dependent contribution to the growth rate. This careful treatment shows that in this class of models, there exists a phase transition between these phases depending on noise and immigration rates.The technical calculations are straight forward and seem to correct (these are mostly in the appendix). The major finding of the paper is a transition from what they call an egalitarian regime (\β>2 in Equation 4) to a dominant regime (\β <2) in the SAD distribution. The authors then fit various SAD curves for four experimental systems in Figure 3.Overall, I found the paper quite compelling and thoughtful. I thought the problem was well motivated and the arguments were well made.I do have some questions and comments that I think could make the paper a little stronger.1. I would like much more discussion and motivation about the difference between \begin{document}$\tilde{\sigma_e}$\end{document} (defined on top of page 3 below Equation 2) and the phenomenological parameter \begin{document}$\sigma_{e}$\end{document} in Equation 3. I am not sure what is exactly being assumed here and why the different notation?

We have added a couple of sentences explaining the origins of \begin{document}$\sigma_{e}$\end{document} and how it differs from the bare amplitude of the environmental noise \begin{document}${\widetilde{\sigma}}_{e}$\end{document} (see discussion around Eq. 5).

2. The authors use a very particular definition of egalitarian and dominance based on the number of species needed to explain half the biomass and how it scales with the size of the ecosystem S? As a statistical physicist, I am wondering if alternative definitions of these quantities such as the Inverse participation Ratio (IPR) also show similar scaling?

A discussion of this point was added at the end of section IIIb

3. The authors make a cryptic statement about seeing only the tails of the ocean prokaryotic data. I would like much more discussion of this.

We have added the following sentence: “Weak sampling shifts the distribution leftward, thus hiding the true characteristics of the SAD for small abundance populations” with a reference to a previous work of our where the effects of weak sampling were analyzed, justifying the above comment.

4. Also, given the very close values of the empirical fits of \β to the critical \β_c=2, it would be very useful to have some sense of error bars? For example, is \β=2.03 really over the transition? I would like some bootstrapped error bars for these fit parameters.

All the relevant details regarding the fit are now listed in Table S1 in the supplementary material.

5. It would be nice to see some simulations for systems that are not strictly neutral and how well these can be approximated by the time-dependent neutral model. How large does the LV interaction matrix (A in Equation. 7) have to be before the arguments here break down for a simulated community?

We thank the reviewer for raising this important point. In the revised version we have added an entire section (IV) devoted to the discussion of this issue.

I think my primary substantial concern is about error bars on the fits. This is quite important for drawing conclusions about the real communities.

We thank the reviewer for raising this important point. As mentioned above, the parameters of the fits for the four ecological data sets are now presented in Table 1 in the SI.